# GraphFormers: GNN-nested Transformers for Representation Learning on Textual Graph

**Junhan Yang**[♦][*], **Zheng Liu**[♣], **Shitao Xiao**[♠], **Chaozhuo Li**[♣], **Defu Lian**[♦],
**Sanjay Agrawal**[♥], **Amit Singh**[♥], **Guangzhong Sun**[♦], **Xing Xie**[♣]
♦ University of Science and Technology of China, Hefei, China
♣ Microsoft Research Asia, Beijing, China
♠ Beijing University of Posts and Telecommunications, Beijing, China
♥ Microsoft India Development Center, Bengaluru, India
`yangjun2@mail.ustc.edu.cn`,
`{zhengliu,cli,siamit,xingx}@microsoft.com`,
`stxiao@bupt.edu.cn`,
`{liandefu,gzsun}@ustc.edu.cn`,
`sanjayiitk0@gmail.com`

## Abstract

The representation learning on textual graph is to generate low-dimensional embeddings for the nodes based on the individual textual features and the neighbourhood information. Recent breakthroughs on pretrained language models and graph neural networks push forward the development of corresponding techniques. The existing works mainly rely on the cascaded model architecture: the textual features of nodes are independently encoded by language models at first; the textual embeddings are aggregated by graph neural networks afterwards. However, the above architecture is limited due to the independent modeling of textual features. In this work, we propose GraphFormers, where layerwise GNN components are nested alongside the transformer blocks of language models. With the proposed architecture, the text encoding and the graph aggregation are fused into an iterative workflow, making each node's semantic accurately comprehended from the global perspective. In addition, a progressive learning strategy is introduced, where the model is successively trained on manipulated data and original data to reinforce its capability of integrating information on graph. Extensive evaluations are conducted on three large-scale benchmark datasets, where GraphFormers outperform the SOTA baselines with comparable running efficiency. The source code is released at `https://github.com/microsoft/GraphFormers` .

## 1 Introduction

The textual graph is a widely existed data format, where each node is annotated with its textual feature. The representation learning on textual graph is to generate low-dimensional node embeddings based on the individual textual features and the information from the neighbourhood. In recent years, the breakthroughs in pretrained language models and graph neural networks contribute to the development of corresponding techniques. Particularly, with pretrained language models, such as BERT (Devlin et al., 2018) and RoBERTa (Liu et al., 2019a), the underlying semantics of texts can be captured more precisely; at the same time, with graph neural networks, like GraphSage (Hamilton et al., 2017a) and GAT (Veličković et al., 2018), neighbours can be effectively aggregated for more informative node embeddings. It is necessary to combine both techniques for better textual graph representation. As

---

[*]Work was done by 2021.01 during Junhan Yang and Shitao Xiao's internship in MSRA

35th Conference on Neural Information Processing Systems (NeurIPS 2021).

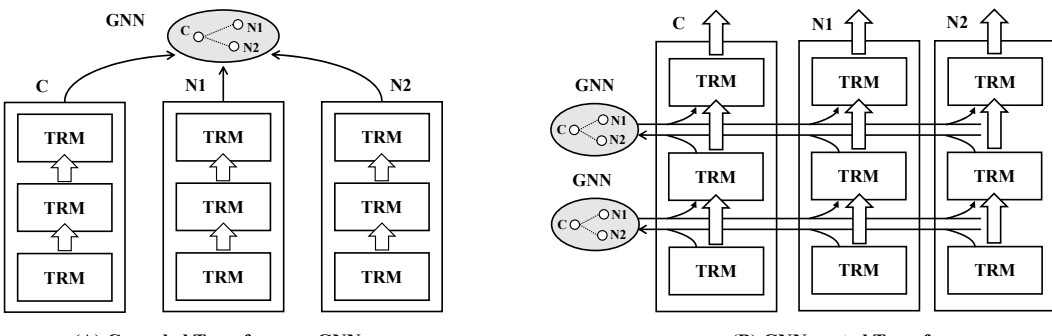

**(A) Cascaded Transformers-GNN**  **(B) GNN-nested Transformers**

Figure 1: Model architecture comparison (a center node C is connected with two neighbours N1, N2). (A) Cascaded Transformers-GNN: text embeddings are independently generated by language models and aggregated by rear-mounted GNNs. (B) GNN-nested Transformers: the text encoding and graph aggregation are iteratively performed with the layerwise GNNs and Transformers (TRM).

suggested by GraphSage (Hamilton et al., 2017a) and PinSage (Ying et al., 2018), the textual feature can be independently modeled by text encoders and further aggregated by rear-mounted GNNs for the final node embeddings. Such a representation paradigm has been widely adopted by subsequent works on various scenarios (Zhu et al., 2021; Li et al., 2021; Hu et al., 2020; Liu et al., 2019b; Zhou et al., 2019), where GNNs are combined with powerful PLM-based text encoders.

The above way of combination is called the "Cascaded Transformers-GNN" architecture (Figure 1 A), as the language models (built upon Transformers) are deployed ahead of the GNN component. With the above architecture, the text encoding and the graph aggregation are performed in two consecutive steps, where there is no information exchange between the nodes when text embeddings are generated. However, the above workflow is defective considering that the linked nodes are correlated, whose underlying semantics can be mutually enhanced. For example, given a node "notes on transformers" and its neighbour "tutorials on machine translation"; by making reference to the whole context, the "transformers" here can be interpreted as a machine learning model, rather than an electric device.

**Our Work**. We propose "GNN-nested Transformers" (**GraphFormers**), which are highlighted for the fusion of GNNs and language models (Figure 1 B). In GraphFormers, the GNN components are nested alongside the transformer layers (TRM) of language models, where the text encoding and graph aggregation are fused as an iteratively workflow. In each iteration, the linked nodes will exchange information with each other in the layerwise GNN component; thus, each node will be augmented by its neighbourhood information. The transformer component will work on the augmented node features, where increasingly informative node representations can be generated for the next iteration. Compared with the cascaded architecture, GraphFormers achieve more sufficient utilization of the cross-node information on graph, which significantly benefit the representation quality. Given that the layerwise GNN components merely involve simple and effective multi-head attention, GraphFormers preserve comparable running costs as the existing cascaded Transformers-GNN models.

On top of the proposed model architecture, we further improve GraphFormers' representation quality and practicability as follows. Firstly, the training of GraphFormers is likely to be shortcut: in many cases, the center node itself can be "sufficiently informative", where the training tasks can be accomplished without leveraging the neighbourhood information. As such, GraphFormers may end up with insufficiently trained GNNs. Inspired by recent success of curriculum learning (Bengio et al., 2009), we propose to **train the model progressively**: the first round of training is performed with manipulated data, where the nodes are randomly polluted; thus, it becomes harder to make prediction merely rely on the center nodes, and the model will be forced to leverage the whole input nodes. The second round of training gets back to the unpolluted data, where the model will be fit into the targeted distribution. Another concern about GraphFormers is that all the linked nodes are mutually dependent in the representation process: once a new node is presented, all the neighbours, regardless of whether they have been processed before, need to be encoded from scratch. As a result, a great deal of unnecessary computations will be incurred. We introduce **unidirectional graph attention** to alleviate this problem: only the center node is required to make reference to the neighbours, while the neighbour nodes remain independently encoded. By this means, the existing neighbours' encoding results can be cached and reused, which significantly saves the computation cost.

Extensive evaluations are conducted with three million-scale textual graph datasets: DBLP, Wiki and Product, where the representation quality is measured by the link prediction accuracy. According to our experiment results, GraphFormers significantly outperform the SOTA cascaded Transformers-GNN baselines with comparable running efficiency.

## 2  Related Work

The textual graph representation is an important research topic in multiple areas, such as natural language processing, information retrieval and graph learning (Yang et al., 2015; Wang et al., 2016b,a; Yasunaga et al., 2017; Wang et al., 2019a; Xu et al., 2019). To learn high-quality representation for textual graph, techniques on natural language understanding and graph representation need to be jointly leveraged. In recent years, breakthroughs on pretrained language models (PLM) and graph neural networks (GNN) significantly advance the development of corresponding techniques.

**PLM**. The PLMs are proposed to learn universal language models with neural networks trained on large-scale corpus. The early works were based on shallow networks, e.g, word embeddings learned by Skip-Gram (Mikolov et al., 2013) and GloVe (Pennington et al., 2014). In recent years, the backbone networks are being quickly scaled up: from EMLo (Peters et al., 2018), GPT (Radford et al., 2018), to BERT (Devlin et al., 2018), XLNet (Yang et al., 2019), T5 (Raffel et al., 2019), GPT-3 (Brown et al., 2020). The large-scale models, which get fully trained with massive data, demonstrate superior performances on general NLP tasks. One of the most critical usages of PLMs is text representation, where the underlying semantics of texts are captured by low-dimensional embeddings. Such embeddings achieve competitive results on downstream tasks, like text retrieval and classification (Reimers and Gurevych, 2019; Luan et al., 2020; Gao et al., 2021; Su et al., 2021).

**GNN**. Graph neural networks are recognized as powerful tools of modeling graph data (Hamilton et al., 2017b; Zhou et al., 2020). Such methods (e.g., GCN (Kipf and Welling, 2016), GAT (Veličković et al., 2018), GraphSage (Hamilton et al., 2017a)) learn effective message passing mechanisms such that information between the nodes can get aggregated for expressive graph representations.

Graph neural networks may also incorporate node attributes, like texts; and it's quite straightforward to leverage GNNs and PLMs for textual graph representation following the "cascaded architecture" suggested by GraphSage (Hamilton et al., 2017a): the node features are independently encoded at first; then, the node embeddings are aggregated via GNNs to generate the final representations. Such a representation paradigm is widely adopted by subsequent works (Zhu et al., 2021; Li et al., 2021; Hu et al., 2020; Liu et al., 2019b; Zhou et al., 2019). However, the above approaches treat the text encoding and graph aggregation as two consecutive steps, where the node-level features are independently processed. Our work is different from these approaches as the text encoding and graph aggregation are fused as an iterative workflow based on the "GNN-nested Transformers".

## 3  GraphFormers

In this work, we deal with textual graph data, where each node $x$ is a text. The node $x$ together with its neighbours $N_x$ are denoted as $G_x$. Our model learns the embedding for node $x$ based on its own textual feature and the information of its neighbourhood $N_x$. The generated embeddings are expected to capture the relationship between the nodes, i.e., to accurately predict whether two nodes $x_q$ and $x_k$ are connected based on the embedding similarity.

### 3.1  GNN-nested Transformers

The encoding process of GraphFormers is indicated as follows. The input nodes (the center node and its neighbours) are tokenized into sequences of tokens, with special tokens [CLS] padded in the front, whose states are used for node representation. The input sequences are mapped into the initial embedding sequences $\{\mathbf{H}_g^0\}_G$ based on the summation of word embeddings and position embeddings. The embedding sequences are encoded by multiple layers of GNN-nested Transformers (shown as Figure 2), where the graph aggregation and text encoding are iteratively performed.

• **Graph Aggregation in GNN**. Each node is enhanced by its neighbourhood information based on the layerwise graph aggregation. For each node in the $l$-th layer, the first token-level embedding (corresponding to [CLS]) is taken as the node-level embedding: $\mathbf{z}_g^l \leftarrow \mathbf{H}_g^l[0]$. The node-level embeddings

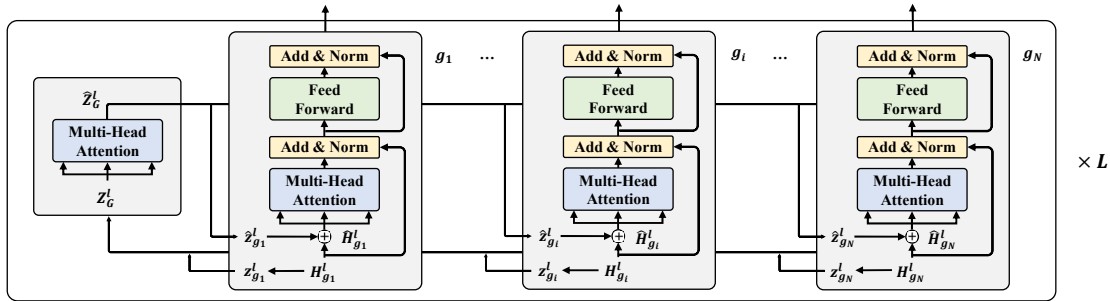

Figure 2: GNN-nested Transformers (using the $l$-th layer for illustration). The graph aggregation is performed in the first place: the node-level embeddings $\{\mathbf{z}_g^l\}_G$ are gathered from all the nodes and processed by the GNN component (the leftmost rectangle). The GNN processed node-level embeddings $\{\hat{\mathbf{z}}_g^l\}_G$ are dispatched to their original nodes, which forms the graph-augmented token-level embeddings. The graph-augmented token-level embeddings are further encoded by Transformer.

are gathered from all the nodes and passed to the layerwise GNN for graph aggregation. We leverage Multi-Head Attention (MHA) to encode the node-level embeddings $\mathbf{Z}_G^l$ ($\{\mathbf{z}_g^l\}_G$), similar as GAT (Veličković et al., 2018). For each attention head, the scaled dot-product attention is performed as:

$$
\begin{aligned}
\hat{\mathbf{Z}}_G^l &= \mathrm{MHA}(\mathbf{Z}_G^l); \\
\mathrm{MHA}(\mathbf{Z}_G^l) &= \mathrm{Concat}(\mathbf{head}_1, ..., \mathbf{head}_h); \\
\mathbf{head}_j &= \mathrm{softmax}(\frac{\mathbf{Q}\mathbf{K^T}}{\sqrt{d}} + \mathbf{B})\mathbf{V}; \\
\mathbf{Q} &= \mathbf{Z}_G^l\mathbf{W}_j^Q; \ \mathbf{K} = \mathbf{Z}_G^l\mathbf{W}_j^K; \ \mathbf{V} = \mathbf{Z}_G^l\mathbf{W}_j^V;
\end{aligned}
\tag{1}
$$

In the above equations, $\mathbf{W}_j^Q$, $\mathbf{W}_j^K$, and $\mathbf{W}_j^V$ are the projection matrices of MHA, corresponding to the $j$-th attention head. A learnable position bias $\mathbf{B}$ is added to the dot-product result; the positions differentiate the relationship between the nodes; i.e., "center-to-center" ($x$ to $x$), "center-to-neighbour" ($x$ to $N_x$), and "neighbour-to-neighbour" ($N_x$ to $N_x$), respectively.

Each of the embeddings $\hat{\mathbf{z}}_g^l$ ($\hat{\mathbf{z}}_g^l \in \hat{\mathbf{Z}}_G^l$) is dispatched to its original node and concatenated ($\oplus$) with the token-level embeddings, which gives rise to the graph-augmented token-level embeddings:

$$
\widehat{\mathbf{H}}_g^l \leftarrow \mathrm{Concat}(\hat{\mathbf{z}}_g^l, \mathbf{H}_g^l).
\tag{2}
$$

In this place, the GNN-processed node-level embeddings $\hat{\mathbf{Z}}_G^l$ can be interpreted as "messengers", with which the neighbourhood information can be introduced to each of the nodes.

• **Text Encoding in Transformer**. The graph-augmented token-level embeddings $\widehat{\mathbf{H}}_g^l$ are processed by the transformer component (Vaswani et al., 2017), where the following computations are performed:

$$
\begin{aligned}
\widehat{\mathbf{H}}_g^l &= \mathrm{LN}(\mathbf{H}_g^l + \mathrm{MHA}^{asy}(\widehat{\mathbf{H}}_g^l)); \\
\mathbf{H}_g^{l+1} &= \mathrm{LN}(\widehat{\mathbf{H}}_g^l + \mathrm{MLP}(\widehat{\mathbf{H}}_g^l)).
\end{aligned}
\tag{3}
$$

In the above equations, MLP is the Multi-Layer Projection unit, and LN is the Layer-Norm unit. We use asymmetric Multi-Head Attention ($\mathrm{MHA}^{asy}$), where $\mathbf{Q}$, $\mathbf{K}$, $\mathbf{V}$ are computed as:

$$
\mathbf{Q} = \mathbf{H}_g^l\mathbf{W}_j^Q; \ \mathbf{K} = \widehat{\mathbf{H}}_g^l\mathbf{W}_j^K; \ \mathbf{V} = \widehat{\mathbf{H}}_g^l\mathbf{W}_j^V.
\tag{4}
$$

Therefore, the output sequence $\mathbf{H}_g^{l+1}$ will be of the same length as the input sequence $\mathbf{H}_g^l$. The encoding result will be used as the input token-level embeddings for the next layer. The node-level embedding at the last layer $\mathbf{z}_x^L$ (i.e., $\mathbf{H}_g^L[0]$) will be used as the final node representation.

• **Workflow**. We summarize GraphFormers' encoding workflow as Algorithm 1. The initial token-level embeddings $\{\mathbf{H}_g^0\}_G$ are independently encoded by the first Transformer layer $\mathrm{TRM}^0$. For a $L$-layer GraphFormers, the graph aggregation and text encoding are iteratively performed for the subsequent $L$-1 steps (from 1 to $L-1$). In each step, the node-level embeddings $\mathbf{Z}_G^l$ are gathered and

---
**Algorithm 1:** GraphFormers' Workflow
---
**Input:** The input graphs $G$ (consist of the center node $x$ and its neighbours).
**Output:** The embedding for the center node $\mathbf{h}_x$.

**1 begin**

**2**     **for** *each text $g \in G$* **do**

**3**        $\mathbf{H}_g^1 \leftarrow \mathrm{TRM}^0(\mathbf{H}_g^0)$; // Get the initial token-level embeddings

**4**     **for** $l = 1, ..., L-1$ **do**

**5**        $\mathbf{Z}_G^l \leftarrow \{\mathbf{z}_g^l | g \in G\}$; // Gather node-level embeddings to GNN

**6**        $\hat{\mathbf{Z}}_G^l \leftarrow \mathrm{GNN}(\mathbf{Z}_G^l)$; // Graph aggregation in GNN

**7**        **for** *each text $g \in G$* **do**

**8**           $\widehat{\mathbf{H}}_g^l \leftarrow \mathrm{Concat}(\hat{\mathbf{z}}_g^l, \mathbf{H}_g^l)$; // Get graph-augmented token-level embeddings

**9**           $\mathbf{H}_g^{l+1} \leftarrow \mathrm{TRM}^l(\widehat{\mathbf{H}}_g^l)$; // Text encoding in Transformer

**10**     Return $\mathbf{h}_x \leftarrow \mathbf{z}_x^L$;

---

processed by the layerwise GNN component. The output node-level embeddings $\hat{\mathbf{Z}}_G^l$ are dispatched to their original nodes, which generates the graph-augmented token-level embeddings $\widehat{\mathbf{H}}_g^l$. The graph-augmented token-level embedding are further processed by the Transformer component. Finally, The node-level embedding (for the center node $x$) in the last layer $\mathbf{z}_x^L$ is taken as our representation result.

• **Encoding Complexity**. Given an input of $M$ nodes, each one has $P$ tokens; the time complexity of each layer's encoding operation is $O(M^2 + MP^2)$: the graph aggregation takes $O(M^2)$, because $M$ node-level embeddings are gathered for multi-head attention; the text encoding takes $O(MP^2)$, as each of the $M$ node calls for the multi-head attention of $P$ tokens. Compared with Transformers, the GNN's computation cost is much smaller, mainly because of two reasons: 1) $M^2 \ll MP^2$ in general, 2) operations like MLP are not needed in graph aggregation. Therefore, the working efficiency of GraphFormers is close to the cascaded GNN-Transformers as the extra computation cost of layerwise graph aggregation is relatively small. Such a property is also empirically verified in our experiment.

### 3.2 Model Simplification: Unidirectional Graph Aggregation

One concern about GraphFormers is that the input nodes are mutually dependent on each other during the encoding process. As a result, to generate the embedding for a node, all the related nodes in its neighbourhood need to be encoded from scratch, regardless of whether they have been processed before. Such a property is unfavorable in practice as a great deal of unnecessary computation cost might be incurred (i.e., a node will be repetitively encoded every time it serves as a neighbour node). We leverage a simple but effective simplification, the unidirectional graph aggregation, to address this problem. Particularly, only the center node $x$ is required to make reference to the neighbourhood; while the rest of nodes $N_x$ remain independently encoded all by their own textual features:

$$\mathbf{H}_g^{l+1} = \begin{cases} \mathrm{TRM}^l(\widehat{\mathbf{H}_x^l}), \ g = x; \\ \mathrm{TRM}^l(\mathbf{H}_g^l), \ \forall g \in N_x. \end{cases} \tag{5}$$

Because the encoding of the neighbour nodes is independent of the center node, the intermediate encoding results $\{\mathbf{z}_g^{1 \cdots L}\}_{N_x}$ can be cached in storage[2] and reused in subsequent computations when they are needed. As a result, the nodes can be prevented from being encoded repetitively, which saves a great deal of unnecessary computation cost. We empirically verify that GraphFormers maintain similar performances when the above simplification is introduced.

### 3.3 Model Training: Two-Stage Progressive Learning

• **Training Objective**. We take advantage of link prediction as our training task. Given a pair of nodes $q$ and $k$, the model is learned to predict whether they are connected based on their embedding

---

[2]The encoding results can be kept in low-cost devices, whose storage capacity can be regarded as infinite.

Table 1: Specifications of the experimental datasets: the number of items, the number of neighbour nodes on average, and the number of training, validation, testing cases.

|        | Product      | DBLP       | Wiki      |
|--------|--------------|------------|-----------|
| #Item  | 5,643,688    | 4,894,081  | 4,818,679 |
| #N     | 4.71         | 9.31       | 8.86      |
| #Train | 22,146,934   | 3,009,506  | 7,145,834 |
| #Valid | 30,000       | 60,000     | 66,167    |
| #Test  | 306,742      | 100,000    | 100,000   |

similarity. Particularly, the following classification loss is minimized for a positive pair of $q$ and $k$:[3]

$$\mathcal{L} = -\log \frac{\exp(\langle \mathbf{h}_q, \mathbf{h}_k \rangle)}{\exp(\langle \mathbf{h}_q, \mathbf{h}_k \rangle) + \sum_{r \in R} \exp(\langle \mathbf{h}_q, \mathbf{h}_r \rangle)}. \tag{6}$$

In the above equation, $\mathbf{h}_q$ and $\mathbf{h}_k$ are the node embeddings; $\langle \cdot \rangle$ denotes the computation of inner product; $R$ stands for the negative samples. In our implementation, we leverage "in-batch negative samples" (Karpukhin et al., 2020; Luan et al., 2020) for the reduction of encoding cost: a positive sample in one training instance will be used as a negative sample in the rest of the training instances within the same mini-batch.

● **Two-stage Training**. In GraphFormers, the information from the center node and neighbour nodes are not treated equally, which may undermine the model's training effect. Particularly, the center node's information can be directly utilized, while the neighbourhood information needs to be introduced via three steps: 1) encoded as node-level embeddings, 2) making graph aggregation with the center node, and 3) introduced to center node's graph augmented token-level embeddings. The message passing pathway can shortcut when the center nodes are "sufficiently informative", i.e., two nodes are sufficiently similar with each other in terms of their own textual features, such that their connection can be predicted without considering the neighbours. Given the existence of such cases, GraphFormers may end up with well-trained Transformers but insufficiently trained GNNs.

To alleviate the above problem, we introduce a warm-up training task, where the link prediction is made based on the polluted input nodes. Particularly, for each input node $g$, a subset of its tokens $g_m$ will be randomly masked[4]. As a result, the classification loss becomes:

$$\mathcal{L}' = -\log \frac{\exp(\langle \mathbf{h}_{\tilde{q}}, \mathbf{h}_{\tilde{k}} \rangle)}{\exp(\langle \mathbf{h}_{\tilde{q}}, \mathbf{h}_{\tilde{k}} \rangle) + \sum_{r \in R} \exp(\langle \mathbf{h}_{\tilde{q}}, \mathbf{h}_{\tilde{r}} \rangle)}, \tag{7}$$

where $\mathbf{h}_{\tilde{q}}$, $\mathbf{h}_{\tilde{k}}$, $\mathbf{h}_{\tilde{r}}$ are the embeddings generated from the polluted nodes. The masked tokens reduce the informativeness of each individual node; therefore, the model is forced to leverage the whole input nodes to make the right prediction.

Finally, the model training is organized as a two-stage progressive learning process. In the first stage, the model is trained to minimize $\mathcal{L}'$ based on the polluted nodes until its convergence, which reinforce the model's capability of integrating information on graph. In the second stage, the model is continually trained to minimize $\mathcal{L}$ based on the original data until the convergence, which makes the model fit into the target distribution.

## 4 Experimental Studies

### 4.1 Data and Settings

We make use of the following three real-world textual graph datasets for our experimental studies.

● **DBLP**[5], which contains the paper citation graph from DBLP up to 2020-04-09. Two papers are linked if one is cited by the other one. The paper's title is used as the textual feature.

● **Wikidata5M**[6] (Wiki) (Wang et al., 2019b), which contains the entity graph from Wikipedia. The first sentence in each entity's introduction is taken as its textual feature.

---

[3] We remove the naive cases where $q$ and $k$ are included by each other's neighbour set, $N_q$ and $N_k$.

[4] We use the common MLM strategy, where 15% of the input tokens are masked: 80% of them are replaced by [MASK], the rest ones are replaced randomly or kept as the original tokens with the same probabilities.

[5] https://originalstatic.aminer.cn/misc/dblp.v12.7z

[6] https://deepgraphlearning.github.io/project/wikidata5m

Table 2: Overall evaluation (GraphFormers marked in bold, the best baseline underlined). Graph-Formers outperforms all baselines, especially the ones based on cascaded Transformers-GNN.

| Methods | Product | | | DBLP | | | Wiki | | |
|---|---|---|---|---|---|---|---|---|---|
| | P@1 | NDCG | MRR | P@1 | NDCG | MRR | P@1 | NDCG | MRR |
| PLM | 0.6563 | 0.7911 | 0.7344 | 0.5673 | 0.7484 | 0.6777 | 0.3466 | 0.5799 | 0.4712 |
| TNVE | 0.4618 | 0.6204 | 0.5364 | 0.2978 | 0.5295 | 0.4163 | 0.1786 | 0.4274 | 0.2933 |
| IFTN | 0.5233 | 0.6740 | 0.5982 | 0.3691 | 0.5798 | 0.4773 | 0.1838 | 0.4276 | 0.2945 |
| PLM+GAT | 0.7540 | 0.8637 | 0.8232 | 0.6633 | 0.8204 | 0.7667 | 0.3006 | 0.5430 | 0.4270 |
| PLM+Max | 0.7570 | 0.8678 | 0.8280 | 0.6934 | 0.8386 | 0.7900 | 0.3712 | 0.6071 | 0.5022 |
| PLM+Mean | 0.7550 | 0.8671 | 0.8271 | 0.6896 | 0.8359 | 0.7866 | 0.3664 | 0.6037 | 0.4980 |
| PLM+Att | 0.7513 | 0.8652 | 0.8246 | 0.6910 | 0.8366 | 0.7875 | 0.3709 | 0.6067 | 0.5018 |
| GraphFormers | **0.7786** | **0.8793** | **0.8430** | **0.7267** | **0.8565** | **0.8133** | **0.3952** | **0.6230** | **0.5220** |

• **Product Graph** (Product), an even larger dataset of online products collected by a world-wide search engine. In this dataset, the users' web browsing behaviors are tracked for the targeted product webpages (e.g., Amazon webpages of Nike shoes). The user's continuously browsed webpages within a short period of time (e.g., 30 minutes) is called a "*session*". The products within a common session are connected in the graph (which is a common way of graph construction in e-commerce scenarios (Ying et al., 2018; Wang et al., 2018)). Each product has its unique textual description, which specifies information like the product name, brand, and saler, etc.

The textual features of all the datasets are in English. We make use of uncased WordPiece (Wu et al., 2016) to tokenize the input text. In our experiment, each text is associated with 5 uniformly sampled neighbours (without replacement); for texts with neighbourhood smaller than 5, all the neighbours will be utilized. We summarized the specifications of all the datasets with Table 1. The experiment results are evaluated in terms of link prediction accuracy, i.e., to predict whether a query node and key node are connected given the textual features of themselves and their neighbours. In each testing instance, one query is provided with 300 keys: 1 positive plus 299 randomly sampled negative cases. We leverage three common metrics to measure the prediction accuracy: **Precision@1**, **NDCG**, and **MRR**. Without specifications, we will take the **unidirectional-simplified GraphFormers** trained with the **two-stage progressive learning** as our default model. More details about the implementations and the training/testing configurations are summarized in an Appendix file. It is submitted together with our source code within the supplementary materials.

## 4.2   Baselines

We focus on the comparison between GNN-nested Transformers and Cascaded Transformers-GNN. To make sure the difference between both architectures can be truthfully reflected from the evaluation results, GraphFormers and the Cascaded Transformers-GNN baselines are equipped with text encoders and graph aggregators of the same capacities. Particularly, we use the BERT-like **PLM** as our text encoder, where UniLM-base[7] (Bao et al., 2020) is chosen as the network backbone for all related methods; the final layer's [CLS] token embedding is used for the text embedding.

We enumerate the following representative graph aggregators as used in GAT (Veličković et al., 2018), GIN (Xu et al., 2018), GraphSage (Hamilton et al., 2017a). The **GAT** aggregator, where the node embedding is generated as the weighted sum of all the text embeddings. Each text embedding's relative importance is calculated as the attention score with the center node. The **Pooling-and-Concat** aggregators, where the center node's text embedding is concatenated with the neighbours' pooling result and linearly transformed for the final representation. Depending on the form of pooling function, we have the following options: **Max** and **Mean**, where neighbours are aggregated by max-pooling and mean-pooling, respectively; **Att**, where the neighbours are summed up based on the attention weights with the center node. By comparison, the neighbourhood information may get more emphasized with GAT; while the center node itself tends to be highlighted with Pooling-and-Concat.

We consider two more baselines which make use of simplified text encoders (such as CNN) and network embeddings: **TNVE** (Wang et al., 2019a) and **IFTN** (Xu et al., 2019). We also include the **PLM** only baseline, which merely leverages the textual feature of the center node.

---

[7]An enhanced BERT-like PLM showing more competitive performances than peers like RoBERTa, XLNet.

Table 3: Impact of neighbour size (#N).

| | GraphFormers | | | PLM+Max | | |
|---|---|---|---|---|---|---|
| #N | P@1 | NDCG | MRR | P@1 | NDCG | MRR |
| 1 | 0.6485 | 0.8087 | 0.7522 | 0.6249 | 0.7946 | 0.7342 |
| 2 | 0.6841 | 0.8308 | 0.7804 | 0.6538 | 0.8137 | 0.7583 |
| 3 | 0.6980 | 0.8396 | 0.7916 | 0.6728 | 0.8256 | 0.7734 |
| 4 | 0.7126 | 0.8485 | 0.8029 | 0.6823 | 0.8319 | 0.7814 |
| 5 | 0.7267 | 0.8565 | 0.8133 | 0.6934 | 0.8386 | 0.7900 |

## 4.3 Overall Evaluation

The overall evaluation results are reported in Table 2. It's observed that GraphFormers consistently outperform all the baselines, especially the ones based on the cascaded Transformers-GNN, with notable advantages. Particularly, it achieves 2.9%, 4.8%, 6.5% relative improvements over the most competitive baselines (underlined) on each of the experimental datasets. Such an observation indicates that the relationship between the nodes can be captured more accurately based on the node embeddings generated by GraphFormers, which verifies the effectiveness of our proposed method.

We also observe the following underlying factors that may influence the representation quality.

Firstly, the effective utilization of neighbourhood information is critical. With the joint consideration of the center node and neighbour nodes, the PLM+GNNs methods, including GraphFormers and the cascaded Transformers-GNN baselines, significantly outperform the PLM only baseline in most of the time. We further analyze the impact of neighbourhood size as Table 3, with a fraction of neighbour nodes randomly sampled for each center node (using DBLP for illustration). It can be observed that both GraphFormers and PLM+Max (the most competitive baseline) achieve higher prediction accuracy than the PLM only method (P@1:0.5673, NDCG:0.7484, MRR:0.6777, as reported in Table 2), even with fewer neighbour nodes included. With the increasing number of neighbour nodes, the advantages become gradually enlarged. However, the marginal gain is vanishing, as the relative improvement becomes smaller when more neighbours are included. In all the testing cases, GraphFormers maintain consistent advantages over PLM+Max, which reaffirms the effectiveness of our proposed methods.

Secondly, the capacity of the text encoder is crucial for textual graph representation. All the pretrained language model based methods (GraphFormers, Cascaded Transformers-GNN baselines, PLM-only baseline) significantly outperform the baselines with simplified text encoders (TNVE, IFTN).

Thirdly, the representation quality is also sensitive to the form of graph aggregator. In Product, the cascaded Transformers-GNN baselines' performances are quite close to each other. In DBLP, PLM+(Max, Mean, Att) outperforms PLM+GAT. In Wiki, not only PLM+(Max, Mean, Att) but also PLM-only baseline outperform PLM+GAT. Such phenomenons could be attributed to the type of graph: whether it is homogeneous or heterogeneous. Particularly, both Product and DBLP can be regarded as homogeneous graphs as the nodes are connected based on the same relationships; i.e., co-view relationship in Product, and citation relationship in DBLP. In both homogeneous graphs, the connected nodes may have quite similar semantics (the co-viewed products usually serve similar user intents, and the citation relationships usually indicate similar research topics); thus, the incorporated neighbour nodes will probably provide complementary information for the link prediction between the center nodes. However, Wiki is a heterogeneous graph, where the connections between entities may have highly different semantics. As a result, the incorporation of neighbour nodes may not contribute to the link prediction task, especially when the incorporated neighbours and the prediction target are connected to the center nodes with totally different relationships. Considering that GAT tends to focus more on the neighbourhood, its performance can be vulnerable in such unfavorable situations. These findings suggest that the neighbourhood information should be properly handled in case that the information of the center node is wiped out.

Finally, we may conclude different methods' utility in textual graph representation: *simplified text encoders ≺ PLMs ≺ Cascaded Transformers-GNN ≺ GNNs-nested Transformers*. Such findings are consistent with our expectation that the precise modeling of individual textual feature and the effective integration of neighbourhood information will jointly contribute to high-quality textual graph representation. GraphFormers enjoy the high expressiveness of PLMs and leverage layerwise nested-GNNs to facilitate graph aggregation, which contributes to both of the above perspectives.

Table 4: Ablation Studies (The top ablated methods are marked in bold; "↑"/"↓": the performance is increased/decreased compared with the default setting). "-Progressive": two-stage progressive learning disabled; "-Simplified": unidirectional simplification disabled; "-Shared GNNs": GNNs parameters are not shared across the layers; "-Position": GNNs learnable position bias disabled.

| Methods | Product | | | DBLP | | | Wiki | | |
|---|---|---|---|---|---|---|---|---|---|
| | P@1 | NDCG | MRR | P@1 | NDCG | MRR | P@1 | NDCG | MRR |
| GraphFormers | 0.7786 | 0.8793 | 0.8430 | 0.7267 | 0.8565 | 0.8133 | 0.3952 | 0.6230 | 0.5220 |
| PLM+Max | 0.7570 | 0.8678 | 0.8280 | 0.6934 | 0.8386 | 0.7900 | 0.3712 | 0.6071 | 0.5022 |
| - Progressive | 0.7688 | 0.8751 | 0.8373 | 0.7096 | 0.8468 | 0.8007 | 0.3834 | 0.6155 | 0.5127 |
| - Simplified | **0.7795** ↑ | **0.8798** ↑ | **0.8436** ↑ | 0.7225 | 0.8542 | 0.8102 | 0.3923 | 0.6209 | 0.5195 |
| - Shared GNNs | 0.7788 | 0.8795 | 0.8433 | 0.7256 | 0.8558 | 0.8123 | **0.3945** ↓ | **0.6221** ↓ | **0.5211** ↓ |
| - Position | 0.7788 | 0.8795 | 0.8434 | **0.7276** ↑ | **0.8570** ↑ | **0.8139** ↑ | 0.3942 | 0.6222 | 0.5211 |

Table 5: Time and memory costs per mini-batch for PLM+Max and GraphFormers, with neighbour size increased from 3 to 200. GraphFormers achieve similar efficiency and scalability as PLM+Max.

| #N | 3 | 5 | 10 | 20 | 50 | 100 | 200 |
|---|---|---|---|---|---|---|---|
| Time: PLM+Max | 60.29 ms | 93.41 ms | 161.40 ms | 295.92 ms | 684.16 ms | 1357.93 ms | 2706.35 ms |
| Time: GraphFormers | 63.95 ms | 97.19 ms | 170.16 ms | 306.12 ms | 714.32 ms | 1411.09 ms | 2801.67 ms |
| Mem: PLM+Max | 1.33 GiB | 1.39 GiB | 1.55 GiB | 1.82 GiB | 2.67 GiB | 4.09 GiB | 6.92 GiB |
| Mem: GraphFormers | 1.33 GiB | 1.39 GiB | 1.55 GiB | 1.83 GiB | 2.70 GiB | 4.28 GiB | 7.33 GiB |

## 4.4 Ablation Studies

The ablation studies (as Table 4) are performed to clarify the following issues: 1) the impact of two-stage progressive learning, and 2) the impact of unidirectional-simplified GraphFormers.

Firstly, the two-stage progressive learning substantially improves GraphFormers' representation quality. Without such a training strategy ("-Progressive": training directly on the original data), the model's performance is decreased by 0.98%, 1.71%, and 1.18% in each of the datasets, respectively.

Secondly, the performances between simplified and non-simplified ("-Simplified") GraphFormers are comparable. In fact, the necessity of graph aggregation is not equivalent for the center node and the neighbour nodes: since the center node is the one for representation, it is much more important to ensure that the center node may extract complementary information from its neighbours. The unidirectional-simplified GraphFormers maintain such a property; thus, there is little impact on the final performances. Such a finding affirms that we may safely leverage the simplified model to save the cost of repetitively encoding the existing neighbours.

We make two additional ablation studies. "-Shared GNNs": the GNNs parameters sharing is disabled, where each layer maintains its own graph aggregator (by default, the layerwise GNN components in GraphFormers share the same set of parameters). "-Position": the learnable position bias (**b** in Eq. 1) is disabled in GNNs. We find that model's performance is little affected from the above changes.

## 4.5 Efficiency Analysis

We compare the time efficiency between GNN-nested Transformers (GraphFormers) and Cascaded Transformers+GNN (using PLM+Max for comparison). The evaluation is made with a Nvidia P100 GPU. Each mini-batch contains 32 encoding instances; each instance contains one center and #N neighbour nodes; the token length of each node is 16. We report the average time and memory (GPU RAM) costs per mini-batch as Table 5.

Firstly, the time and memory costs of both methods grow linearly with the increment of neighbours. (There are overheads of time and memory costs. The time cost overhead may come from CPU processing; while the memory cost overhead is mainly due to the model parameters (Rajbhandari et al., 2020)). We may approximately remove the overheads by deducting the time and memory costs where #N=3). Such a finding is consistent with our theoretical analysis in Section 3.1.

Secondly, the overall time and memory costs of GraphFormers are quite close to PLM+Max. When the number of neighbour nodes is small, the differences between both methods are almost ignorable. The differences become slightly larger when more neighbour nodes are included, because the layerwise

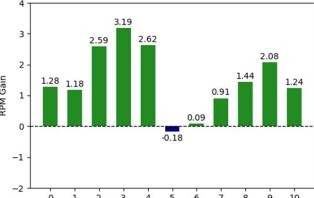 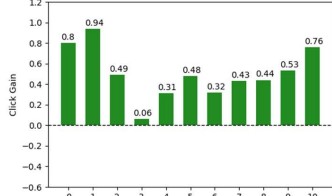 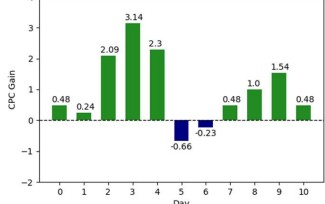

Figure 3: Online A/B Test: the relative improvements of RPM, CY and CPC against the last version of production system in Bing Search (green: positive; blue: negative). In most of the time, all three performance indicators are significantly improved thanks to the utilization of GraphFormers.

graph aggregations in GraphFormers get increasingly time consuming. However, the differences are still relatively small: merely around 3.5% of the overall running costs when #N is increased to 200 ("#N=200" is already more than enough for most of the real world scenarios).

Based on the above observations, we may conclude that GraphFormers are more accurate, meanwhile equally efficient and scalable as the conventional cascaded Transformer+GNNs.

### 4.6 Online A/B Test on Bing Search

GraphFormers has been deployed as one of the major ads retrieval algorithms on Bing Search, and it achieves highly competitive performance against the previous production system (the combination of a wide spectrum of semantic representation algorithms, including large-scale PLMs and cascaded PLMs-GNNs). Particularly, the primary objective of Ads service is to maximize the revenue meanwhile increasing the user clicks. Therefore, the following three metrics are taken as the major performance indicators: RPM[8] (revenue per thousand impressions), CY (click yield), and CPC[9] (cost per click) . During our large-scale online A/B test, GraphFormers significantly improves the overall RPM, CY, CPC by 1.87%, 0.96% and 0.91%, respectively. A 11-day performance snapshot is demonstrated as Figure 3; it can be observed that in most of the time, all three metrics are significantly improved thanks to the utilization of GraphFormers (the daily performance are measured based on millions of impressions, thus having strong statistic significance).

## 5 Conclusion

In this paper, we propose a novel model architecture GraphFormers for textual graph representation. By having GNNs nested alongside each transformer layer of the pretrained language model, the underlying semantic of each textual node can be precisely captured and effectively integrated for high-quality textual graph representation. On top of the fundamental architecture, we introduce the two-stage progressive training strategy to further strengthen GraphFormers' representation quality; we also simplify the model with the unidirectional graph aggregation, which eliminates the unnecessary computation cost. The experimental studies on three large-scale textual graph datasets verify the effectiveness of our proposed methods, where GraphFormers notably outperform the existing cascaded Transformer-GNNs methods with comparable running efficiency and scalability.

## 6 Acknowledgement

We are grateful to anonymous reviewers for their constructive comments on this work. The work was supported by grants from the National Natural Science Foundation of China (No. 62022077).

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
