# A Implementation Details

## A.1 Masking Strategy

We use span masking (Joshi et al., 2020) as our masking strategy. For each iteration, we sample and mask a span of text, until the ratio of masked tokens has reached the threshold. We follow the settings in (Joshi et al., 2020). The span length $l$ is generated from a geometric distribution $l \sim Geo(p)$, where $p$ is set to 0.2 and $l$ is clipped at $l_{max} = 10$. As in BERT (Devlin et al., 2018), 15% of the input tokens will be masked: 80% of them are replaced by [MASK], 10% are replaced by random tokens and 10% are kept as the original tokens.

## A.2 GraphFormers' Workflow

Algorithm 1 provides the pseudo-code of GraphFormers' workflow. We use original Multi-Head Attention in the first Transformer layer ($\mathrm{Transformers}[0]$), and asymmetric Multi-Head Attention in the rest Transformer layers ($\mathrm{Transformers}[1..L-1]$). In original Multi-Head Attention, $\mathbf{Q}$, $\mathbf{K}$, $\mathbf{V}$ are computed as:

$$\mathbf{Q} = \mathbf{H}_g^l \mathbf{W}_j^Q; \ \mathbf{K} = \mathbf{H}_g^l \mathbf{W}_j^K; \ \mathbf{V} = \mathbf{H}_g^l \mathbf{W}_j^V. \tag{1}$$

In asymmetric Multi-Head Attention, $\mathbf{Q}$, $\mathbf{K}$, $\mathbf{V}$ are computed as:

$$\mathbf{Q} = \mathbf{H}_g^l \mathbf{W}_j^Q; \ \mathbf{K} = \widehat{\mathbf{H}}_g^l \mathbf{W}_j^K; \ \mathbf{V} = \widehat{\mathbf{H}}_g^l \mathbf{W}_j^V. \tag{2}$$

In the above equations, $\mathbf{H}_g^l$ are token-level embeddings, $\widehat{\mathbf{H}}_g^l$ are graph-augmented token-level embeddings, and $\mathbf{W}_j^Q$, $\mathbf{W}_j^K$, and $\mathbf{W}_j^V$ are the projection matrices of Multi-Head Attention, corresponding to the $j$-th attention head.

In each step, we extract the embeddings of [CLS] tokens as node-level embeddings $\mathbf{Z}_g^l$. The node-level embeddings $\mathbf{Z}_g^l$ and a learnable bias vector $\mathbf{b}$ are processed by the GNN component, which is a Multi-Head Attention layer. The output GNN-processed node-level embeddings $\hat{\mathbf{Z}}_g^l$ are concatenated with the original token-level embeddings $\mathbf{H}_g^l$, which generates the graph-augmented token-level embeddings $\widehat{\mathbf{H}}_g^l$. Then $\widehat{\mathbf{H}}_g^l$ are processed by the Transformer component using asymmetric Multi-Head Attention. At last, the node-level embedding of the center node $\mathbf{h}_x$ is returned as the representation of the graph.

# B Training Details

As shown in Table 1, we present the hyperparameters used for training GraphFormers. The model is trained for at most 100 epochs on all datasets. For the stability of the training process, we optimally tune the learning rate as $1e{-}5$ for Product, $1e{-}6$ for DBLP, and $5e{-}6$ for Wiki. We use an early stopping strategy on P@1 with a patience of 2 epochs and Adam (Kingma and Ba, 2014) with $\beta_1$=0.9, $\beta_2$=0.999, $\epsilon$=1e-8 for optimization. We pad the sequence length to 32 for Product and DBLP, 64 for Wiki, depending on different text length of each dataset. To make full use of the GPU memory, we set the batch size as 240 for Product and DBLP, 160 for Wiki. Each training sample includes 12 nodes: 1 query with its 5 neighbours, and 1 keyword with its 5 neighbours. The training is on $8\times$ Nvidia V100-16GB GPU clusters. The training of GraphFormers takes 58.8, 117.6, 151.2 hours on average to converge on each of the experimental datasets (Product, DBLP, Wiki). We use Python3.6 and PyTorch 1.6.0 for implementation. The random seeds of PyTorch and NumPy are fixed as 42. For two-stage training, the training processes of the two stages share the same settings as above.

**Algorithm 1:** GraphFormers' Workflow in PyTorch-Like Style

```
# Input:
# Hg[0]: initial token-level embeddings (summation of word embeddings and position
    embeddings)
# Output:
# hx: output embeddings

# B: batch size
# N: number of nodes in the graph (0th node represents the center node)
# SL: sequence length
# D: hidden dimension
# L: number of GNN-nested Transformer layers
# b: learnable bias vector for nodes

# token-level embeddings: BxNxSLxD
Hg[1] = Transformers[0](Hg[0].view(B * N, SL, D), asymmetric = False).view(B, N, SL,
    D) # "asymmetric = False" means we use original Multi-Head Attention in the
    Transformer

for l in range(1, L):

    # node-level embeddings: BxNxD
    Zg[l] = Hg[l][:, :, 0]

    # GNN-processed node-level embeddings: BxNxD
    Zg_hat[l] = MultiHeadAttention(Zg[l], b)

    # graph-augmented token-level embeddings: BxNx(SL+1)xD
    Hg_hat[l] = Concat([Zg_hat[l][:, :, None, :], Hg[l]], dim = 2)

    # token-level embeddings: BxNxSLxD
    Hg[l + 1] = Transformers[l](Hg_hat[l].view(B * N, SL + 1, D), asymmetric = True).
        view(B, N, SL, D) # "asymmetric = True" means we use asymmetric Multi-Head
        Attention in the Transformer

# graph representations: BxD
hx = Hg[L][:, 0, 0, :]

return hx
```

Table 1: Hyperparameters for training GraphFormers

| Optimizer | Adam | | |
|---|---|---|---|
| Adam $\beta_1$ | 0.9 | | |
| Adam $\beta_2$ | 0.999 | | |
| Adam $\epsilon$ | 1e-8 | | |
| PyTorch random seed | 42 | | |
| NumPy random seed | 42 | | |
| | Product | DBLP | Wiki |
| Max training epochs | 100 | 100 | 100 |
| Learning rate | 1e-5 | 1e-6 | 5e-6 |
| Sequence length | 32 | 32 | 64 |
| Batch size | 240 | 240 | 160 |