# OpenReview forum: "GraphFormers: GNN-nested Transformers for Representation Learning on Textual Graph"
_NeurIPS.cc/2021/Conference — NeurIPS 2021 Poster_

### Official Review · Reviewer_Kwpr · 2021-07-11

**Rating:** 7
**Confidence:** 4

**Summary:**

This paper proposes GraphFormers for text graph tasks where GNNs are nested in each layer of the transformer-based pretrained language model. In this way, each node can appropriate graph structure to build semantic representations more effectively. Two-stage progressive training and unidirectional graph aggregation are further proposed to improve representation quality and avoid unnecessary computation. They use three real-world million-scale textual graph datasets (DBMLP, Wikidata, Product Graph) and demonstrate consistent improvements compared with traditional cascaded transformers-GNN models.

**Limitations And Societal Impact:**

The authors have acknowledged the limitations of their current work including (1) not including edge information, (2) only used for textual graphs. This opens up further directions.

I don't see any potential negative societal impact.


**Main Review:**

Originality: The author first identify the weakness of the "cascaded transformers-GNN" architecture: the text encoding and graph aggregation are in two consecutive with no information exchanged in text embeddings. Based on this insight, the proposed approach differs from the traditional "cascaded transformers-GNN" by interleaving transformers and GNNs. They have adequately discussed a comprehensive set of related work in pretrained LM and GNNs.

Quality: While this modification is simple, it is reasonable to nest GNNs into transformers to better encode information incorporating graph structures. The example in line 37-40 is helpful to illustrate the usefulness. The early usage and repetitive usage of graph structure is the main advantage of GraphFormers versus traditional manners. The added nested GNN operation is also small compared to transformers as discussed algorithmically and empirically.

In experiments, through comparison with other SOTA models and different variants of PLM+GNNs, the advantages of GraphFormers are supported and verified.

However, the authors should clarify two things:
1. Clearly define the center nodes against the neighbour nodes. Which node is defined as center nodes in each tasks? and how this will affect the results?
2. In each layer of GNN, is the parameter shared? In other words, do you make sure the number of parameters is the same between GNN-nested transformers and Cascaded Transformers-GNN?

Clarify: The paper is well written and easy to follow. The technical components are documented in detail with equations and visualizations. However, the center node should be clearly defined.

Significance: In general, I believe their results of this new technique are solid. While the technique is simple and straightforward, comprehensive results have been presented to valid its effectiveness and efficiency. The proposed method can potentially change how other researchers use PLM and GNNs in a wide range of NLP tasks.


**Time Spent Reviewing:**

3

---

> ### Author Response · Authors · 2021-08-10
> **Response to Reviewer Kwpr**
>
> Thanks a lot for your acknowledgement and insightful comments! We would like to make the following clarifications w.r.t. your concerns
>
> **Comment.**Clearly define the center nodes against the neighbour nodes. Which node is defined as center nodes in each tasks? and how this will affect the results?
>
> **Response.** A general definition of the center/neighbour node is made as follows. Our model is for the representation of textual graph, where each node denotes a document, and each edge denotes the connection between two documents (e.g., the linkage relationship between two Wikipedia articles). Our mode is trained with the link-prediction task: to predict the existence of an edge. Therefore, given a sampled edge, each of the edge’s endpoints is called a “center node”; meanwhile, the nodes connected to the center nodes (except the center nodes themselves) are called “neighbour nodes”. More specifically, for DBLP dataset: if a paper is chosen as the center node, then the papers within its reference are called the neighbour nodes; for Wiki dataset: if a Wiki article is chosen as the center node, then its linked articles are called the neighbour nodes; for Product dataset: if a product is chosen as center node, then its co-clicked products are called the neighbour nodes.
> The differentiation of center and neighbour nodes is critical, because the center node is relatively more important for the link prediction task. We empirically verify this point by making one more experimental study, where “GraphFormers (no diff)” indicates the method without the differentiation of center and neighbour nodes (we use bi-directional GraphFormers, where we take the average of all the nodes’ final node-level embeddings for the graph representation, such that all nodes are treated equally). Similarly, we introduce “PLM+Max (no diff)” (PLM+Max is the strongest PLM+GNN baseline in the experiment), where the final graph representation is generated by taking the average of PLM+Max embeddings for all nodes (run PLM+Max for multiple times, by taking each of the input nodes as center; and then take the average of all nodes’ PLM+Max embeddings for the graph representation).
> We find that the performances drop notably for both GraphFormers (no diff) and PLM+Max (no diff), indicating that the differentiation of center and neighbour is critical for the representation quality.
>
> |                        | P@1    | NDCG   | MRR    |
> |------------------------|--------|--------|--------|
> | PLM+Max                | 0.6934 | 0.8386 | 0.7900 |
> | PLM+Max (no diff)      | 0.6857 | 0.8335 | 0.7836 |
> | GraphFormers           | 0.7225 | 0.8542 | 0.8102 |
> | GraphFormers (no diff) | 0.7021 | 0.8415 | 0.7942 |
>
> **Comment.** In each layer of GNN, is the parameter shared? In other words, do you make sure the number of parameters is the same between GNN-nested transformers and Cascaded Transformers-GNN?
>
> **Response.** The GNN parameters are shared across different layers by default; that is to say, we maintain the same scale of parameters as the cascaded Transformers-GNN. According to our ablation study in Table 4 (the comparison between “GraphFormers” and “- Shared GNNs”), the parameter sharing does not bring any negative effect to GraphFormers’ performance.

---

### Official Review · Reviewer_1iRC · 2021-07-16

**Rating:** 6
**Confidence:** 4

**Summary:**

This paper proposes a means to encode text feature for nodes in textual graphs, where a layerwise aggregation module (implemented as a multi-head self-attention) is appended after each Transformer encoder block that aggregates the hidden representations of (sampled) neighbor nodes. The produced contextual representations are dispatched to each node for the computation of next Transformer encoder block. To mitigate the possible model degradation issue that the aggregation module might be insufficiently trained, the authors also propose a two-stage training schedule where the model is first trained on nodes with corrupted inputs until convergence and then trained on normal data.

**Limitations And Societal Impact:**

In the last section, the authors list some of the limitations of the proposed method: 1. it doesn't take the edge information into account; 2. It is specifically designed for textual graphs.



**Main Review:**

This paper proposes a novel text feature extraction method for textual graphs where intermediate representations of each node are contextualized using representations from neighbor nodes. It also proposes a two-stage training schedule to balance the capacity of the pre-trained text encoder and the aggregation modules.

The motivation that node representations should be contextualized in order to distinguish between nodes with similar texts but very different semantics is interesting. And the main method in this paper is clearly represented overall. However, there are still some concerns that I'd prefer the authors to address:

- Position bias: In Eq. (1),  a learnable position bias is added to the unnormalized attention score where three types of bias (c2n,c2c,n2n) are utilized. What is the purpose of the bias term? It doesn't seem important to distinguish between the center node and the neighbor nodes. The ablation result seems to be insignificant and is not elaborated.
- On data settings: In DBLP citation graph, the text feature is chosen to be the title of each paper, which I suspect might be not informative enough to encode the semantics of the paper. This issue is especially crucial when the titles are short. I was wondering if the authors have considered using abstracts as well.
- Performance on Wiki: In line 274, the authors provide justification for the inferior performance of PLM+GAT on Wiki (see Table 2) from the heterogeneity of Wiki graph and that the incorporation of neighbor nodes might not contribute to the link prediction task. This justification seems at odds with the motivation and the performance of the proposed model (i.e., that the neighbor nodes can serve as additional information in generating the node representations).

Overall I think the paper is built upon good motivations, but some aspects of the empirical results are not explained or not convincing enough.

( Several minor notation issues and typos:

- In Eq. (1), a multi-head self-attention is used to aggregate the information between nodes, in which a learnable position bias $\mathbf{b}$ is added to the attention score. I believe the notation should be $\mathbf{B}$ by convention.
- Other typos: line 188: leverage (what?) of; line 257: varnishing -> vanishing; line 267: be attribute(d) to

)



**Time Spent Reviewing:**

30

---

> ### Author Response · Authors · 2021-08-10
> **Response to Reviewer 1iRC**
>
> Thanks so much for the constructive comments! We appreciate the acknowledgement of our motivation, model design and training method. We find several concerns are raised w.r.t. the position bias, data settings and performance on Wiki. In this place, we make the following clarifications and experimental studies to address them.
>
> **Comment.** Position bias: In Eq. (1), a learnable position bias is added to the unnormalized attention score where three types of bias (c2n,c2c,n2n) are utilized. What is the purpose of the bias term? It doesn't seem important to distinguish between the center node and the neighbor nodes. The ablation result seems to be insignificant and is not elaborated.
>
> **Response.** 1. Purpose of bias term. When we perform graph aggregation, there are two kinds of information we can leverage: 1) the semantic information, which is reflected by the textual content of each node, 2) the topological information, which is reflected by the positions of nodes. Recent studies, like [A] and [B], show that both kinds of information are useful in learning the representations for general graphs. Inspired by these works, we introduce the position bias to our GNN module, which helps to encode the topological relationship between the nodes (i.e., center-to-center, center-to-neighbour, neighbour-to-neighbour). Intuitively, it is reasonable to assume that the center-to-center and center-to-neighbour attention should be emphasized with higher attention weights, because they are more directly related to the generation of center node’s representation. We believe this design is beneficial to the model expressiveness, as the whole available information (semantic and topology) can be jointly leveraged to compute the attention weights for graph aggregation.
> 2. Experimental insight. Despite the enhanced expressiveness, our experiment shows that the model benefits little from such a component as you mentioned. One explanation is that: since we are dealing with textual graphs (where the textual contents are the dominating features), the semantic information may greatly outweigh the topological information. Therefore, the attention scores can be precisely computed purely based on the node-level embeddings (which are generated on top of the textual contents). We believe this empirical finding, which goes against the recognition in general graphs, will provide insight for the future works on learning textual graph representations. We will make this point more explicit in our experiment analysis.
>
> [A] Position-aware Graph Neural Networks, https://arxiv.org/pdf/1906.04817.pdf
> [B] Do Transformers Really Perform Bad for Graph Representation? https://arxiv.org/pdf/2106.05234.pdf
>
> **Comment.** On data settings: In DBLP citation graph, the text feature is chosen to be the title of each paper, which I suspect might be not informative enough to encode the semantics of the paper. This issue is especially crucial when the titles are short. I was wondering if the authors have considered using abstracts as well.
>
> **Response.** “Using the title only” is a common setting in the existing works studying the representation of citation graphs (e.g., [C], [D]). We believe this is an empirically reasonable trade-off between training efficiency and final performance; therefore, we inherit this setting in our work. However, it is still interesting to study the impact of using both title and abstract as suggested, in case that some titles are too short to provide sufficient information. In this place, we perform additional experiments for the analysis.
>
> |                               	| P@1    	| NDCG   	| MRR    	|
> |-------------------------------	|--------	|--------	|--------	|
> | PLM (title)                   	| 0.5673 	| 0.7484 	| 0.6777 	|
> | PLM+Max (title)               	| 0.6934 	| 0.8386 	| 0.7900 	|
> | GraphFormers (title)          	| 0.7267 	| 0.8565 	| 0.8133 	|
> | PLM (title+abstract)          	| 0.6757 	| 0.8238 	| 0.7719 	|
> | PLM+Max (title+abstract)      	| 0.7223 	| 0.8553 	| 0.8114 	|
> | GraphFormers (title+abstract) 	| 0.7543 	| 0.8723 	| 0.8336 	|
>
> We make the following observations from the experiment results. Firstly, the performances of all the methods (baselines and GraphFormers) are consistently improved when the title and abstract are jointly utilized. This observation is in our expectation, because the abstract will bring additional information, which benefits the representation quality. Secondly, GraphFormers maintain notable advantages over PLM+Max (the strongest PLM+GNN baseline), while PLM+Max still dominates the PLM only baseline. This observation further verifies our conclusion: 1) the neighbour nodes bring additional information, which benefits the representation of textual graph; 2) GraphFormers have stronger capability of utilizing neighbour nodes, therefore leading to better performance compared with the conventional cascaded PLM+GNNs methods.
>
> [C] Heterogeneous Graph Transformer, https://arxiv.org/pdf/2003.01332.pdf
> [D] GPT-GNN: Generative Pre-Training of Graph Neural Networks, https://arxiv.org/pdf/2006.15437.pdf

---

> > ### Author Response · Authors · 2021-08-10
> > **More responses for additional comments.**
> >
> > **Comment.** Performance on Wiki: In line 274, the authors provide justification for the inferior performance of PLM+GAT on Wiki (see Table 2) from the heterogeneity of Wiki graph and that the incorporation of neighbor nodes might not contribute to the link prediction task. This justification seems at odds with the motivation and the performance of the proposed model (i.e., that the neighbor nodes can serve as additional information in generating the node representations).
> >
> > **Response.** Indeed, it is very interesting to see that PLM+GAT performs poorly on Wiki (we confirm this finding by double checking the implementation and the running of the experiments). We further clarify our explanation and provide additional experiments to justify our analysis.
> > Our major arguments are two-fold: 1) Wiki’s neighbours are noisy, 2) PLM+GAT is vulnerable to noise. Thus, PLM+GAT performs poorly on Wiki; while other methods (including GraphFormers and other PLM+GNNs baselines), which are more robust to noise, may still leverage the useful information within Wiki’s neighbours to improve their performances.
> > On one hand, we argue that the neighbour nodes in Wiki suffer from stronger noise than the other two datasets. That’s to say, there are still informative neighbours in Wiki, which provide useful information for the graph representation; however, the ratio of non-informative neighbours in Wiki can be higher than the other two datasets, which bring stronger background noise. To quantitatively justify the above argument, we introduce the following analysis: we compare the embedding similarities between 1) query’s neighbour to key, 2) query to key’s neighbour, 3) query’s neighbour to key’s neighbour (we use the output embedding of the PLM text encoder, and take the cosine similarity of the embeddings). The higher similarity indicates stronger correlation. And if “one center node’s neighbours” have strong correlation with “the other center node” or “the other center node’s neighbours”, then the introduction of neighbour nodes will benefit the prediction of the linkage between the two center nodes (i.e., the neighbours are informative).
> >
> > |                                      | Product | DBLP   | Wiki   |
> > |--------------------------------------|---------|--------|--------|
> > | query’s neighbour-to-key             | 0.7100  | 0.5004 | 0.4178 |
> > | query-to-key’s neighbour             | 0.7368  | 0.5455 | 0.4394 |
> > | query’s neighbour-to-key’s neighbour | 0.7731  | 0.4966 | 0.3237 |
> >
> > It can be observed from the above result that For Wiki, the “similarity between one center’s neighbours and the other center” (i.e., query’s neighbour-to-key, query-to-key’s neighbour) and the “similarity between the neighbours of the two centers” (i.e., query’s neighbour to key’s neighbour) are much smaller than those on Product and DBLP. That’s to say, the introduced neighbours are less relevant to the link prediction task (by making this statement, we mean the ratio of informative neighbour is lower, rather than all neighbours are non-informative).
> >
> > On the other hand, the PLM+GAT is more vulnerable to background noise due to the lack of “direct linkage between the center node and the graph aggregation result”. In fact, both our GraphFormers and other PLM+GNN baselines have the component where the center node can be directly linked to the graph aggregation result (in GraphFormers: the center node and the aggregated neighbours are added up via residual connection; in PLM+Max/Mean/Att: the center node and the aggregated neighbours are concatenated together). This component is critical for the model’s robustness: when the sampled neighbours are highly noisy in some situations, it may still provide meaningful prediction because the center node information (which is more important than the neighbours in link prediction) can always be fully preserved. To some extent, it makes the model benefit from graph aggregation result when the sampled neighbours are informative; meanwhile, it safeguards the model’s performance when the sampled neighbours are noisy (maintains a comparable accuracy as using the center nodes only). To justify the above analysis, we perform one more experiment, where the “direct linkage”  is introduced to the PLM+GAT baseline: the original PLM+GAT embedding is concatenated with the center node embedding for the final text graph representation. It can be observed that the prediction accuracy of PLM+GAT is significantly improved after adding the direct linkage, which becomes comparable to the other PLM+GNN baseline.
> >
> > |                               | P@1    | NDCG   | MRR    |
> > |-------------------------------|--------|--------|--------|
> > | PLM+GAT                       | 0.3006 | 0.5430 | 0.4270 |
> > | PLM+GAT (with direct linkage) | 0.3690 | 0.6043 | 0.4990 |
> > | PLM+Mean                      | 0.3664 | 0.6037 | 0.4980 |
> > | PLM+Att                       | 0.3709 | 0.6067 | 0.5018 |
> > | PLM+Max                       | 0.3712 | 0.6071 | 0.5022 |
> > | GraphFormers                  | 0.3952 | 0.6230 | 0.5220 |
> >
> > **Comment.** In Eq. (1), a multi-head self-attention is used to aggregate the information between nodes, in which a learnable position bias b is added to the attention score. I believe the notation should be B by convention.
> > Other typos: line 188: leverage (what?) of; line 257: varnishing -> vanishing; line 267: be attribute(d) to
> >
> > **Response.** Thanks a lot for pointing out these problems, we will revise our presentation based on your suggestions.

---

> > > ### Comment · Reviewer_1iRC · 2021-08-30
> > > **Reading the other reviews and responses, I raised my score.**
> > >
> > > Reading the other reviews and responses, I raised my score.

---

### Official Review · Reviewer_nCrZ · 2021-07-17

**Rating:** 7
**Confidence:** 4

**Summary:**

The paper proposes GraphFormer, an architecture built on top of GAT and Transformers for textual graphs. The motivation behind the architecture is that the existing techniques on textual graphs involve a cascade process where the nodes are first encoded using a language model after which a GNN is applied. With GraphFormers, the graph aggregation and encoding can happen iteratively, allowing for better contextualization of nodes with text like “notes on transformers” since transformers can be interpreted as ML models or electrical devices depending on the surrounding context.

The authors also lay out a 2-step training procedure whereby they first train on a version of the graph with injected noise via link prediction before training on the actual graph. The motivation behind this is given as the fact that training on the graph with injected noise makes the GNN component of the GraphFormer model more robust without which the transformer part will dominate.

**Limitations And Societal Impact:**

The authors do clearly call out the limitation of the work in that GraphFormers are only applicable to textual graphs and does not take edge information like edge type or edge features into account. Therefore, they can’t be applied to heterogenous graphs.

**Main Review:**

Overall, this paper is a good one.  I like the motivation presented by the authors for their work. Indeed the cascade process for encoding textual graphs is not optimal, and the community as a whole will benefit from having a more concerted approach.

Strengths of the paper:
* The presented architecture is intuitive to understand where GNN and transformer are iteratively applied in the neighborhood aggregation process to allow for text at a node to be encoded taking into account the graph context.
* Several experiments are performed to show the superiority of GraphFormer over cascaded approaches in link prediction task on three benchmark datasets.
* Authors utilize caching in a succinct manner to decrease the recomputation needed for encoding different nodes in the graph; the caching works by preventing nodes that are neighbors of a central node to aggregate from the neighbors.

Weaknesses of the paper:
* The organization of the paper and the way in which the equations for GraphFormer are presented make the paper a bit hard to follow.
* There are other works on making GNNs and aggregations within them scalable via caching (e.g., https://arxiv.org/pdf/2001.07524.pdf) but the authors have not made use of that, instead reinventing the wheel a little.

**Time Spent Reviewing:**

2.5 hours

---

> ### Author Response · Authors · 2021-08-10
> **Response to Reviewer nCrZ**
>
> We would like to thank the reviewer for the acknowledgement and constructive comments! We make the following responses to the given suggestions.
>
> **Comment.** The organization of the paper and the way in which the equations for GraphFormer are presented make the paper a bit hard to follow.
> **Response.** We will revise the presentation accordingly, by introducing explicit references between the equations and their related components in GraphFormers.
>
> **Comment.** There are other works on making GNNs and aggregations within them scalable via caching (e.g., https://arxiv.org/pdf/2001.07524.pdf) but the authors have not made use of that, instead reinventing the wheel a little.
> **Response.** Thanks for the suggestion, we will try our best to reuse the existing tools in our future implementation.

---

> > ### Comment · Reviewer_nCrZ · 2021-08-11
> > **Responding to authors' response**
> >
> > Reading the reviews and responses, I am keeping my score.
> >
> > > Thanks for the suggestion, we will try our best to reuse the existing tools in our future implementation.
> >
> > That would be great, but at least in this work, the authors should cite previous works like https://arxiv.org/pdf/2001.07524.pdf on caching in GNNs to indicate that the concept is not novel.

---

> > > ### Author Response · Authors · 2021-08-12
> > > **Second round of response to Reviewer nCrZ**
> > >
> > > Thanks a lot for your advice, we will add discussion in the current manuscript w.r.t. the related works on caching in GNNs.

---

### Official Review · Reviewer_yPsh · 2021-07-18

**Rating:** 6
**Confidence:** 3

**Summary:**

For representation learning on textual graphs, Cascaded Transformers-GNN architectures have been used, as language models and GNN are independently and consecutively deployed. The authors' claims, however, this approach has a limitation of information fusion of text encoding and graph structures. The authors proposed GNN-nested Transformers named as GraphFormers, where GNN components are nested alongside the transformer layers of language models. Also, the author proposed a progressive learning strategy and unidirectional graph attention to effectively train the graph network modules. The efficacy of the GraphFormers is demonstrated on 3 textual graph datasets (DBLP, Wiki, and Product Ads datasets), achieving SOTA performance.

**Ethical Concerns:**

.

**Limitations And Societal Impact:**

The authors addressed the limitations of this work in terms of the problem scopes. From the author's checklist, I could find the Product Graph dataset is a private industrial dataset. I would like to recommend denote this and that the dataset will not be published due to the privacy issue in the main manuscript.

**Main Review:**

1. Originality
    1. The problem this paper tackle is interesting in which utilizing the neighborhood information during text encoding, and the proposing approach that inserting GNN layers into the transformer layers of PLM seems novel.
    2. However, the nested GNN module's functionality is utilizing the related information of neighborhood nodes to learn the center node's representation using multi-headed attention layers. My main concern is that the proposed model should be theoretically and experimentally explained the differences to feeding all concatenated texts (i.e., $[g_1; g_2; ... ; g_n] , g_i \in G_x$)  into transformer-based PLMs (especially large scale PLMs such as GPT-x, T5, etc) as an input. The concatenated text input approach may have advantages in the sense of more flexible and rich attention representation.
2. Quality
    1. This paper's claims are technically sound, and experimental results support the efficacy of GraphTransformers.
    2. The proposed uni-directional graph attention for model simplification and two-stage progressive learning methods are well-studied through the ablation studies.
3. Clarity
    1. Overall, the paper is well-organized and well-written.
    2. In figure 2, the asymmetric MHA is not correctly represented. From the figure, readers can misunderstand as symmetric MHA of $\hat{H_g^l}$). Furthermore, in Algorithm 1 (in Appendix), Transformers[l]( Hg_hat[l], asymmetric = True) is not understandable. According to eq.4, the MHA^asy requires two different inputs.
4. Significance
    1. Lack of baseline of recent previous work (TwinBERT, AdsGNN, TextGNN, MetaBERT)
    2. Lack of comparison with other PLMs.
        - What is 'BERT-like PLM' exactly?
        - Lack of leveraging recent large-scale PLMs such as T5 and GPT-x.
    3. Lack of **comparison with feeding concatenated text inputs into large scale PLMs**
    4. The Product Ads dataset is not published so that it is hard to reproduce the results. Since the dataset is new, the details such as how the dataset is collected, categories of the items, etc. should be explained.
    5. Lack of qualitative analysis.
    6. From table 4, there was little performance decrease when the learnable position bias is disabled, which concluding the bias is unnecessary.

**Time Spent Reviewing:**

5

---

> ### Author Response · Authors · 2021-08-10
> **Response to Reviewer yPsh**
>
> We would like to thank the reviewer for the acknowledgement and insightful comments! We find several concerns are raised w.r.t. the baselines, experiment settings, dataset settings, etc.  In the place, we address them with the following clarifications and experiments.
>
> **Comment.** However, the nested GNN module's functionality is utilizing the related information of neighborhood nodes to learn the center node's representation using multi-headed attention layers. My main concern is that the proposed model should be theoretically and experimentally explained the differences to feeding all concatenated texts (i.e., [g1;g2;...;gn],gi∈Gx) into transformer-based PLMs (especially large scale PLMs such as GPT-x, T5, etc) as an input. The concatenated text input approach may have advantages in the sense of more flexible and rich attention representation.
> Lack of comparison with feeding concatenated text inputs into large scale PLMs
>
> **Response.** Theoretical and experimental comparison with “feeding all concatenated texts into transformer-based PLMs” (“*all-concatenated-PLMs*” for short).  The all-concatenated-PLMs may serve as a brute-force method for textual graph representation. However, we would like to highlight the following differences in comparison with GraphFormers.
>
> 1. Topology. All-concatenated-PLMs lack for effective utilization of graph topology. By concatenating all texts together, there is no explicit differentiation of whether two token-level embeddings are coming from the same node (it can only do it implicitly, by injecting special tokens in the bottom layer). However, in GraphFormers, there is an explicit boundary between “the intrinsic information from each node itself”, and “the external information from the neighbourhood”. Particularly, the intra-node information is encoded by the token-level embeddings of each node, and the neighbourhood information is introduced by the GNN-processed node-level embeddings. We argue that such explicit boundaries are important. For all-concatenated-PLMs: when the number of neighbour nodes is large, the center node information may get “over attenuated”, given that the center node needs to aggregate with an overwhelming number of tokens from neighbour nodes; while for GraphFormers, the center node’s information may always get highlighted, because the neighbourhood information (which is supplementary and relatively less important than the center node information) is always introduced by one single GNN-processed node-level embedding. We experimentally justify this advantage in the following experimental analysis.
> 2. Scalability. By using Transformers to process the entire concatenated result, the time and memory complexity of all-concatenated-PLMs increase quadratically w.r.t. the overall token length (i.e., O(m^2 \* n^2), where m is the number of nodes and n is the text length of each node). That’s to say, the computation cost will grow dramatically with the increment of neighbour nodes. Besides, the PLMs’ input token length is upper bounded, e.g., 512 for RoBERTa, 300 for T5. Given the above restrictions, it is infeasible for all-concatenated-PLMs to incorporate a large number of neighbour nodes. In contrast, the time and memory complexity of GraphFormers is O(m*n^2), which is much smaller than all-concatenated-PLMs. Meanwhile, the overall input size (the center node together with all neighbour nodes) is not subject to the upper bound of token length, since PLMs process each node individually (only need to make sure one single text is within the token length upper bound). Therefore, we are able to utilize much more neighbour nodes with GraphFormers. It should be noted that high scalability will also contribute to accuracy: the enlarged receptive field (i.e., the increased number of neighbour nodes) will let the model access to more comprehensive information, therefore helping the model to make more accurate prediction.
> 3. Adaptability. The all-concatenated-PLMs lack for adaptability: when the graph is changed, e.g., a new edge is added or an existing edge is deleted, all-concatenated-PLMs need to encode everything from scratch (the concatenation of the center node and its neighbours). This will lead to considerable computation cost, which restricts it from being applied to dynamic graphs (in fact, most of the real-world graphs are dynamic, like Ads and Items in E-commerce platforms, which call for periodic update). With GraphFormers (uni-directional), the model only needs to update the center node (or the newly added nodes), since the encoding results of the existing neighbours can all be reused (as introduced in Section 3.2).
>
> **Experimental analysis.**
> The experimental comparison between all-concatenated-PLMs and GraphFormers is shown as follows (using DBLP for the analysis). For all-concatenated, the number of neighbours is upper bounded by 15 (i.e., N <= 15), because the maximum token length of UniLMv2 is 512 (same value for most of other PLMs, like BERT, RoBERTa), and the text length of each node is 32. As a result, all-concatenate PLMs become infeasible when N>15 (i.e., overall length equals to (1+15)*32). For GraphFormers, we increase the number of neighbours to 15, 20, and 25.
>
>
> || P@1| NDCG| MRR|
> |---|---|---|---|
> |all-concatenated (N=5)|0.7336| 0.8600| 0.8178|
> |all-concatenated (N=15)|0.7520| 0.8712| 0.8322|
> |all-concatenated (N>15)| --| --| --|
> |GraphFormers (N=5)| 0.7267| 0.8565 	| 0.8133|
> |GraphFormers (N=15)| 0.7518| 0.8712| 0.8321|
> |GraphFormers (N=20)| 0.7539| 0.8724| 0.8337|
> |GraphFormers (N=25)| 0.7550| 0.8730| 0.8344|
>
> We make the following observations from the experiment. Firstly, all-concatenated is slightly better than GraphFormers when N=5; this is in our expectation because all-concatenate serves as a brute-force method for text graph representation, where all the nodes may directly get access to the whole contextual information via self-attention. When N=15, the performances of all-concatenated and GraphFormers are improved because of the expanded information from the extra neighbours. Meanwhile, the performances of GraphFormers and all-concatenated become comparable. We would attribute this phenomenon to GraphFormers’ better capability in dealing with graph topology (the 1st point discussed above). Finally, when N becomes even larger to 20 and 25, GraphFormers’ performances are further increased and become even better than all-concatenated PLM (which become infeasible when dealing with anything larger than N=15).
>
>
> **Comment.** Lack of baseline of recent previous work (TwinBERT, AdsGNN, TextGNN, MetaBERT)
> Lack of comparison with other PLMs.
> 1. What is 'BERT-like PLM' exactly?
> 2. Lack of leveraging recent large-scale PLMs such as T5 and GPT-x.
>
> **Response.** We make additional experimental comparisons with TwinBERT, AdsGNN, MetaBERT on DBLP (TextGNN is actually the PLM+GAT baseline in our original experiment).
> The 'BERT-like PLM' is UniLMv2 (as illustrated in line 226 in our paper, Unilmv2: Pseudo-masked language models for unified language model pre-training, Bao et. al., ICML 2020). In fact, UniLMv2 is a more recent model than T5  and GPT, and it shows stronger performances than T5 and GPT in many downstream tasks. In this place, we also study the impact of using different PLMs by leveraging T5, GPT-2.
>
> |                      | P@1    | NDCG   | MRR    |
> |----------------------|--------|--------|--------|
> |                      |        |        |        |
> | TwinBERT             | 0.5601 | 0.7465 | 0.6749 |
> | TextGNN              | 0.6633 | 0.8204 | 0.7667 |
> | AdsGNN               | 0.6997 | 0.8417 | 0.7941 |
> | MetaBERT             | 0.6611 | 0.8197 | 0.7656 |
> |                      |        |        |        |
> | PLM+Max              | 0.6934 | 0.8386 | 0.7900 |
> | GraphFormers         | 0.7267 | 0.8565 | 0.8133 |
> |                      |        |        |        |
> | PLM+Max (T5)         | 0.6872 | 0.8345 | 0.7848 |
> | GraphFormers (T5)    | 0.7120 | 0.8477 | 0.8019 |
> |                      |        |        |        |
> | PLM+Max (GPT-2)      | 0.6681 | 0.8237 | 0.7708 |
> | GraphFormers (GPT-2) | 0.6842 | 0.8307 | 0.7804 |
>
> (Without specification, all methods use UniLMv2 PLM for fair comparisons.)
>
> We make the following observations from the experiments. Firstly, GraphFormers dominate all the additional baselines in terms of prediction accuracy (all with UniLMv2 PLM backbone). Secondly, although using different PLM backbones may affect performances, GraphFormers maintain notable advantages over PLM+Max (which is the strongest PLM+GNN baseline in our original experiment) . Thirdly, the GPT-2 backbone is inferior to the other PLMs (T5, UniLMv2). This is because GPT-2 is a uni-directional language model, which is inappropriate for natural language understanding tasks compared with other bi-directional ones, like BERT, RoBERTa, T5, UniLMv2, etc.

---

> > ### Author Response · Authors · 2021-08-10
> > **Add more responses to additional comments**
> >
> > **Comment.** The Product Ads dataset is not published so that it is hard to reproduce the results. Since the dataset is new, the details such as how the dataset is collected, categories of the items, etc. should be explained.
> >
> > **Response.** The Product Ads dataset is a large-scale dataset in addition to the other two public ones, based on which we may evaluate GraphFormers’ effectiveness in real-world ads production systems (we believe this will benefit researchers and practitioners from industry). This dataset is provided by the advertisement team from one of the major world-wide search engines. We make further clarification in addition to our introduction in the original paper (line 202--208). In this dataset, each node is an advertisement webpage of one product. The nodes are connected based on users’ co-click behaviors: if two products are clicked by the same user within a short period of time (a.k.a. a session, which is set to 30 min in our production system), then the corresponding nodes are connected by an edge. The items are not limited to specific categories; they can be anything related to online advertisements, e.g., food, clothes, vehicles, electronic devices, etc. (There are a vast number of advertisers with hundreds of millions of products, where our dataset samples a subset of them. There are about 5.6 million items in our dataset as shown in Table 1.)
> >
> > **Comment.** Lack of qualitative analysis.
> >
> > **Response.** We showcase the effectiveness of GraphFormers with the following example from Wiki. (We’ll also release the well-trained checkpoint as we stated in the paper so that readers may generate more cases)
> >
> > ** Query ** (center node)
> > Elisabeth Gantt (born 1934) is a botanist, known for her work in plant physiology and structure.
> > ** Query’s Neighbours **
> > QN1: The United States of America (USA), commonly known as the United States
> > QN2: Humans (Homo sapiens) are the only extant members of the subtribe Hominina
> > QN3: Plant anatomy or phytotomy is the general term for the study of the internal structure of plants
> > QN4: Northwestern University (NU) is a private research university based in Evanston, Illinois, United States, with other campuses located in Chicago and Doha, Qatar, and academic programs and facilities in Miami, Florida; Washington, D.C.; and San Francisco, California
> > ---
> > ** Key ** (center node)
> > The Gilbert Morgan Smith Medal is awarded by the U.S.
> > ** Key’s Neighbours **
> > KN1: Gilbert Morgan Smith (6 January 1885, Beloit, Wisconsin – 11 July 1959) was a botanist, who worked primarily on the algae.
> > KN2: Richard Cawthorn Starr (August 24, 1924 – February 3, 1998) was an American phycologist.
> > KN3: Ruth Sager (February 7, 1918 – March 29, 1997) was an American geneticist.
> > KN4: Shirley Winifred Jeffrey  (4 April 1930 – 4 January 2014) was an Australian marine biologist and naturalist, who researched   biochemical separation techniques, specialising in micro-algal research; her discovery, isolation and purification of chlorophyll c allowed for the evaluation of oceanic microscopic plant biomass and photosynthesis.
> >
> > For this example, GraphFormers predict the right key for the given query, while all the PLM+GNN baselines and PLM only baseline make wrong predictions. To make right prediction, the model needs to comprehend the underlying semantic of “The Gilbert Morgan Smith Medal is awarded by the U.S.”. From the key’s center node, we may only get to know that “Gilbert Morgan Smith Medal” can be some award in USA. However, when we jointly encode the key’s center node with its neighbour nodes (with our GNN-nested Transformers which enable the neighbourhood-aware comprehension), we may infer that “Gilbert Morgan Smith Medal” can be an award for botanist, geneticist, biologist, etc. Since “Elisabeth Gantt” is a botanist according to the textual description of query, the model will be able to figure out the connection between the query and key. This example demonstrates that 1) the neighbour nodes contribute to the quality of text graph representation, and 2) GraphFormers achieve more effective utilization of the neighbours’ information.
> >
> > **Comment.** From table 4, there was little performance decrease when the learnable position bias is disabled, which concluding the bias is unnecessary.
> >
> > **Response.** 1. Purpose of bias term. When we perform graph aggregation, there are two kinds of information we can leverage: 1) the semantic information, which is reflected by the textual content of the node, 2) the topological information, which is reflected by the positions of nodes. Recent studies, like [A] and [B], show that both kinds of information are useful in learning the representations for general graphs. Inspired by these works, we introduce the position bias to our GNN component. We believe this design is beneficial to the model expressiveness, as both available information (semantic and topology) can be jointly used to compute the attention weights for graph aggregation.
> > 2. Experimental insight. Despite the enhanced expressiveness, our experiment shows that the model benefits little from such a component as you mentioned. One explanation is that: since we are dealing with ordinary textual graphs (where the textual contents are the dominating features), the semantic information may greatly outweigh the topological information. Therefore, the attention scores can be truthfully computed purely based on the node-level embeddings (which are generated from the content features). We believe this empirical finding, which goes against the recognition in general graphs, will provide insight for the future works on learning textual graph representations. We will make this point more explicit in our experiment analysis.
> > [A] Position-aware Graph Neural Networks, https://arxiv.org/pdf/1906.04817.pdf
> > [B] Do Transformers Really Perform Bad for Graph Representation? https://arxiv.org/pdf/2106.05234.pdf
> >
> > **Comment.** In figure 2, the asymmetric MHA is not correctly represented. From the figure, readers can misunderstand as symmetric MHA of). Furthermore, in Algorithm 1 (in Appendix), Transformers[l]( Hg_hat[l], asymmetric = True) is not understandable. According to eq.4, the MHA^asy requires two different inputs.
> > **Response.** Thanks for the suggestion. In fact, there are some differences between these two MHA, we'll update our figure to highlight this point. For the transformer encoding: Hg_hat[l] is the concatenation of two elements: Zg_hat[l] and Hg[l], while our MHA^asy requires Hg_hat[l] and Hg[l]. The pseudo code is simplified because Hg[l] is already included in Hg_hat[l]. We'll clarify this point in the paper.

---

> > > ### Author Response · Authors · 2021-08-17
> > > **Adding additional experiment results for the comparison with all-concatenated-PLMs.**
> > >
> > > We extend our experimental analysis of the comparison with all-concatenated-PLMs, by making use of Wiki in addition to DBLP.  In Wiki, the text length is 64; as a result, all-concatenated-PLMs may incorporate at most 7 neighbours (the overall length is upper bounded by 64*(1+7)).
> > >
> > > |                        | P@1    | NDCG   | MRR    |
> > > |------------------------|--------|--------|--------|
> > > | all-concatenated (N=5) | 0.3950 | 0.6226 | 0.5216 |
> > > | GraphFormers (N=5)     | 0.3952 | 0.6230 | 0.5220 |
> > > |                        |        |        |        |
> > > | all-concatenated (N=7) | 0.3990 | 0.6264 | 0.5261 |
> > > | GraphFormers (N=7)     | 0.4016 | 0.6291 | 0.5292 |
> > > |                        |        |        |        |
> > > | GraphFormers (N=10)    | 0.4097 | 0.6367 | 0.5383 |
> > > | GraphFormers (N=15)    | 0.4131 | 0.6397 | 0.5418 |
> > >
> > > We find that when N=5, GraphFormers and all-concatenated achieve comparable performances; when N=7, GraphFormers outperform all-concatenated; and with the introduction of more neighbours, i.e., N=10 and15, GraphFormers' prediction accuracy becomes even higher (notably higher than all-concatenated with its max input size). The overall tendency is similar as what we observe in DBLP; however, GraphFormers' superiority in dealing with massive neighbours is more obvious given the enlarged advantages in accuracy. Our explanation is that for Wiki, the neighbour nodes are more likely to introduce noise (as explained in our response to Reviewer 1iRC); therefore, the differentiation between the center and neighbour nodes is more important than DBLP, which magnifies the impact of topology.

---

> > > > ### Comment · Reviewer_yPsh · 2021-08-26
> > > > **Responding to authors' response**
> > > >
> > > > I carefully read the authors' rebuttal, and thanks for addressing my concerns with the clarifications and extensive experiments. My main concern was comparison with "all-concatenated-PLMs", and the author explained this in terms of topology, scalability, and adaptability and showed experimental supports. Although the "all-concatenated-PLMs" and GraphFormers resulted in almost similar performances, the scalability of the GraphFormer enables it to deal with longer texts and larger neighbors. I'm still not sure that whether the GraphFormer is superior to PLMs with a much larger input token length. Nevertheless, I agree that most of PLMs have 512 token sizes so that the scalability of GraphFormer is important.
> > > >
> > > > Also, I could find the comparisons with other baselines and examples for qualitative analysis that the author provided in the rebuttal. These additional results and explanations should be reflected in the modified manuscript.
> > > >
> > > > I have another question and concern about the Product Ads dataset.
> > > > - Will this dataset be published? Are there any privacy and ethical issues during the collection of this dataset?
> > > > - In line 202, "Product Graph" is required to be corrected as "Product Ads dataset".

---

> > > > > ### Author Response · Authors · 2021-08-30
> > > > > **Response to Reviewer yPsh**
> > > > >
> > > > > Thanks so much for the enlightening response! We will update our paper with the discussions made in the rebuttal stage. Regarding the newly provided comments, we make more clarifications and experimental analysis.
> > > > >
> > > > > For the concerns about the Product Ads dataset:
> > > > > * We find no privacy and ethical issues so far. In fact, this dataset only contains two kinds of information: 1) the textual descriptions about the product ads, and the co-click relationships between the product ads ("co-click" only indicates two product ads have ever been clicked by the same anonymous users). Therefore, no user information is released from this dataset.
> > > > > * We will release this dataset, and we will work closely with our legal and business team to ensure the user's privacy is fully protected.
> > > > > * We will revise the presentation issue as suggested.
> > > > >
> > > > > For concerns about the performance of larger input sizes. As discussed, the input size of PLMs is upper bounded (no more than 512 for most of the PLMs). To process the input larger than 512, we have to modify the original PLMs such that the length of input tokens may go beyond 512, i.e., by introducing more learnable positions for tokens beyond 512. (However, this modification is not a common practice). We get the following results for the modified PLMs (for DBLP, N=20/25: 672/832 tokens; for Wiki, N=10/15: 704/1024 tokens).
> > > > >
> > > > > |DBLP| P@1| NDCG| MRR|
> > > > > |---|---|---|---|
> > > > > |all-concatenated (N=20)|0.7508| 0.8704| 0.8312|
> > > > > |all-concatenated (N=25)|0.7493| 0.8697| 0.8302|
> > > > > |GraphFormers (N=20)| 0.7539| 0.8724| 0.8337|
> > > > > |GraphFormers (N=25)| **0.7550**|**0.8730**|**0.8344**|
> > > > >
> > > > > |Wiki| P@1| NDCG| MRR|
> > > > > |---|---|---|---|
> > > > > |all-concatenated (N=10)|0.3977| 0.6259| 0.5254|
> > > > > |all-concatenated (N=15)|0.3944| 0.6235| 0.5225|
> > > > > |GraphFormers (N=10)| 0.4097| 0.6367| 0.5383|
> > > > > |GraphFormers (N=15)| **0.4131**| **0.6397**| **0.5418**|
> > > > >
> > > > >
> > > > > We get the following observation from the above results: GraphFormers consistently outperform all-concatenated with notable advantages. We attribute such advantages to GraphFormers' better capability of leveraging graph topology. Such a capability is critical for dealing with large input sizes (discussed in our previous response), as it makes the model steadily benefit from the supplementary information introduced from the enlarged neighbours.
> > > > >
> > > > > To summarize, GraphFormers are not only more scalable and adaptable than all-concatenated, but also achieve higher precision by leveraging large input more effectively.

---

> ### Comment · Reviewer_yPsh · 2021-08-30
> **Update my review**
>
> I carefully read the authors' responses and updated my review score from 5 to 6.
> The authors resolved my main concerns about the GraphFormers' superiority by providing experimental and conceptual comparison with "all-concatenated-PLMs" and other baselines. Furthermore, the authors responsed that the priviate dataset Product Ads has no ethical issues and they will open this dataset to the public.

---

### Author Response · Authors · 2021-08-10
**Acknowledgement.**

We sincerely thank AC for handling this paper and all reviewers' constructive comments! The comments have enlightened us to think deeper and made our work more solid.

We are encouraged by the reviewers' acknowledgements about our motivation, model design and training method. We find a few concerns mainly about the experiments. We address all of them with substantial experiment studies and detailed clarifications, including
1. comparison with all the additional baselines required by the reviewers, together with comprehensive analysis of the experiment results.
2. theoretical and experimental analysis of the original experiment results, which are questioned by the reviewers.
3. questions about the performances under different settings.

We also make clarifications to all the other minor concerns questioned by the reviewers.

---

### Decision · Program_Chairs · 2021-09-27

**Decision:**

Accept (Poster)

**Comment:**

This paper proposes a new model for learning representations on a textual graph. The key idea is to combine Transformer and graph neural network. Experimental results show that the proposed approach works better than the traditional cascade approach.

Strength
* The paper is generally clearly written, although there is still room for improvements on the presentation.
* The proposed model appears to be reasonable and technically sound.
* Experimental results have demonstrated the effectiveness of the proposed method.
* The reviewers pointed out some issues with the paper. The authors have successfully addressed most of them, particularly they provide new experimental results.

Weakness
* There are details that are not explained very clearly. The authors are encouraged to further improve the presentation.
* There are still problems with the English of the paper. For example, “graph aggregation and text encoding are iterative performed” --> it should be "iteratively". The authors are encouraged to do further proofreading. Please significantly revise the paper based on your replies in the rebuttal if the paper is accepted.

Minor comments: It is not clear for non-experts what the downstream tasks are from the explanation in the abstract and introduction. Please add that.